# Genome-scale metabolic models highlight stage-specific differences in essential metabolic pathways in *Trypanosoma cruzi*

Isabel S. Shiratsubaki[1,2☉], Xin Fang[2☉], Rodolpho O. O. Souza[3], Bernhard O. Palsson[2,4,5], Ariel M. Silber[3]*, Jair L. Siqueira-Neto[1]*

**1** Skaggs School of Pharmacy and Pharmaceutical Sciences, UC San Diego, La Jolla, California, United States of America, **2** Department of Bioengineering, UC San Diego, La Jolla, California, United States of America, **3** Laboratory of Biochemistry of Tryps - LaBTryps, Department of Parasitology, Institute of Biomedical Science, University of São Paulo, São Paulo, São Paulo, Brazil, **4** Department of Pediatrics, UC San Diego, La Jolla, California, United States of America, **5** The Novo Nordisk Foundation Center for Biosustainability, Technical University of Denmark, Lyngby, Denmark

☉ These authors contributed equally to this work.
* asilber@usp.br (AMS); jairlage@health.ucsd.edu (JLS-N)

**Data Availability Statement:** The pan-model and stage-specific models have been deposited to the BiGG Models database:

## Abstract

Chagas disease is a neglected tropical disease and a leading cause of heart failure in Latin America caused by a protozoan called *Trypanosoma cruzi*. This parasite presents a complex multi-stage life cycle. Anti-Chagas drugs currently available are limited to benznidazole and nifurtimox, both with severe side effects. Thus, there is a need for alternative and more efficient drugs. Genome-scale metabolic models (GEMs) can accurately predict metabolic capabilities and aid in drug discovery in metabolic genes. This work developed an extended GEM, hereafter referred to as iIS312, of the published and validated *T. cruzi* core metabolism model. From iIS312, we then built three stage-specific models through transcriptomics data integration, and showed that epimastigotes present the most active metabolism among the stages (see S1–S4 GEMs). Stage-specific models predicted significant metabolic differences among stages, including variations in flux distribution in core metabolism. Moreover, the gene essentiality predictions suggest potential drug targets, among which some have been previously proven lethal, including glutamate dehydrogenase, glucokinase and hexokinase. To validate the models, we measured the activity of enzymes in the core metabolism of the parasite at different stages, and showed the results were consistent with model predictions. Our results represent a potential step forward towards the improvement of Chagas disease treatment. To our knowledge, these stage-specific models are the first GEMs built for the stages Amastigote and Trypomastigote. This work is also the first to present an *in silico* GEM comparison among different stages in the *T. cruzi* life cycle.

## Author summary

Chagas disease is a neglected tropical disease that affect millions of people. Chagas disease is caused by *T. cruzi*, a protozoan that can be transmitted through insect vectors.

http://bigg.ucsd.edu/models/iIS312, http://bigg.ucsd.edu/models/iIS312_Trypomastigote, http://bigg.ucsd.edu/models/iIS312_Epimastigote, http://bigg.ucsd.edu/models/iIS312_Amastigote. All other relevant data are within the manuscript and its Supporting Information files.

**Funding:** This research is supported by Microbial Science Initiative Graduate Research Fellowship (UCSD Center for Microbiome Innovation), Novo Nordisk Foundation Center for Biosustainability and the Technical University of Denmark (grant number NNF10CC1016517), Foundation for Research Support of the State of São Paulo (FAPESP) grants 2016 / 06034-2 to AMS and the joint grant to AMS and Prof. Michael Barrett (University of Glasgow) co-funded by MRC and FAPESP 2018/14432-3, Brazil National Council for Scientific and Technological Development (CNPq) grants 301971 / 2017-0 and 404769 / 2018-7, Research Council United Kingdom Global Challenges Research Fund under grant agreement "A Global Network for Neglected Tropical Diseases" (grant MR/P027989/1). The funders had no role in study design, data collection and analysis, decision to publish, or preparation of the manuscript.

**Competing interests:** The authors have declared that no competing interests exist.

Currently, medications for chagas disease are limited and have severe side effects. *T. cruzi* presents a complex multi-stage life cycle including 3 stages, which poses additional challenges in drug development. In this study, we deepened the understanding of the metabolism of *T. cruzi* across its life cycle through integrating omics data with GEMs. We reconstructed an updated GEM of *T.cruzi* with additional subsystems using novel genome annotations—the number of gene included increased by ~50%. We then generated models for each stage using stage-specific transcriptomics data to model only genes that were expressed at each stage. Simulations of stage-specific models suggest remarkable differences in metabolism across stages of *T. cruzi*. Gene and pathway essentiality was also predicted to vary greatly across its multi-stage life cycle, some of which were confirmed by previous studies. Our models were verified with experimentally measured enzyme activity at each stage. In conclusion, our results revealed stage-specific differences in metabolism in *T. cruz* and provided valuable insight into drug discovery.

## Introduction

Chagas disease is a neglected tropical disease affecting millions of people, especially in the Americas where it is endemic [1]. The transmission can occur vertically from an infected mother to baby during gestation, by blood transfusion or organ transplant, through contaminated food, in laboratory accidents, but most commonly by an insect vector that carries the parasite *Trypanosoma cruzi*, a protozoan of the family *Trypanosomatidae*.

Chagas disease presents two clinical phases during infection: acute, and chronic. In the acute phase, when symptoms are present they are typically mild, including fever, fatigue, body aches, headaches, and rashes. The chronic phase is characterized by an undetectable parasitemia and a robust immune response in immunocompetent individuals. Furthermore, it most frequently presents an indeterminate asymptomatic form that can last for decades. However, in some patients it leads to a symptomatic form characterized by cardiac and/or digestive complications, which makes Chagas disease one of the leading causes of heart failure in the Americas. Current anti-Chagas drug options are limited to benznidazole and nifurtimox. Both drugs can cause severe gastrointestinal, dermatological, and neurological side effects, and are more efficient when the infected individual is treated during the acute phase. In the chronic phase, the efficacy of these drugs is unsatisfactory [2,3]. Thus, there is an urgent need for alternative and more efficient drugs.

*T. cruzi* presents a complex multi-stage life cycle that takes place in both vertebrate and invertebrate hosts, and has three main stage-specific forms of the parasite as actors: epimastigotes (replicative and insect-specific), trypomastigotes (non-replicative and infective, bloodstream forms derived from amastigotes and metacyclic forms derived from epimastigotes), and amastigotes (replicative and mostly intracellular in the vertebrate host humans) [4]. Although the three stages were first defined by their morphological characteristics [5], they also present differences at the cellular and biochemical levels, including cell architecture, composition of the surface molecules, and in energy metabolism [4,6,7]. Findings in the literature indicate that epimastigotes have an active metabolism (catabolism and anabolism), and use different nutrients as energy source (lipids, sugars, and amino acids) [8,9]. In contrast, trypomastigotes present low levels of transcription and translation, and since their main function in the *T. cruzi* cycle is infection, they are specialized in attachment and invasion of the host cells [4]. Finally, very little information is available regarding amastigote metabolism.

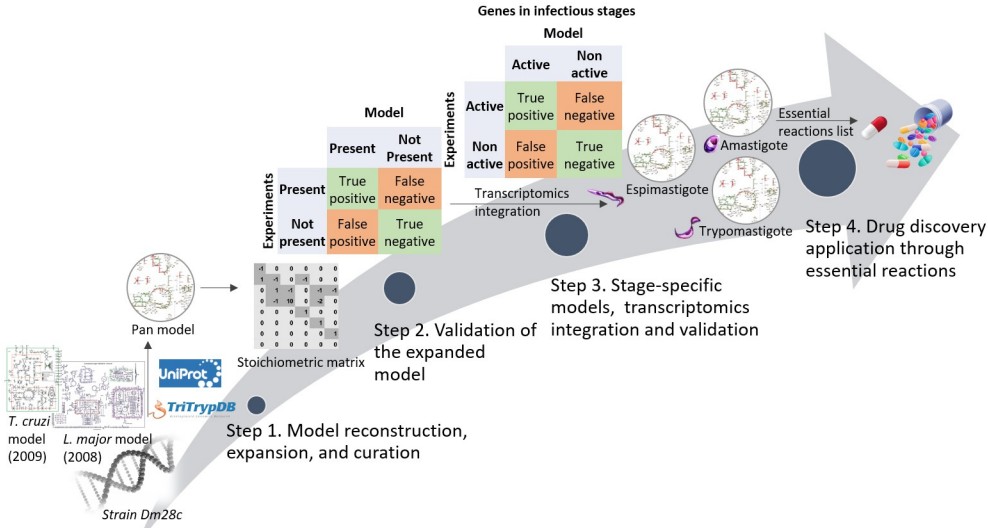

**Fig 1. The workflow for the multi-stage reconstructions of *T. cruzi*.** Step 1. The reconstruction of *T. cruzi* multi-stages used the genome annotation of the strain Dm28c [14], the models iSR215 (*T. cruzi* strain CL Brener) [13] and iAC560 (*L. major* strain Friedlin) [15], the literature, the genomic database for pathogens of the family Trypanosomatidae (TriTryDB) [16], and other genomic databases. Step 2. The expanded model iIS312 was validated using experimental data. Step 3. Stage-specific models (epimastigote, amastigote, and trypomastigote) were generated through transcriptomics integration and validated using experimental data from the infectious stage. Step 4. Analysis of metabolic functions and potential application to drug discovery.

Metabolic genes are good candidates in the identification of potential drug targets, since they are critical for cellular growth and survival [10]. In addition, genome-scale metabolic models (GEMs) [11] have been developed to accurately predict metabolic capabilities from genome sequences. Genome-scale metabolic network reconstructions are knowledge bases that map genotypes to phenotypes. Such reconstructions can then be converted to mathematical models—GEMs—to enable computation of metabolic functions of the organism on a systems level. Flux balance analysis (FBA), one of the most commonly used methods in GEM analysis, is used in this study to investigate the GEMs built. FBA is able to calculate the flux distributions that optimize the objective function (such as biomass function) with the given constraints, including steady-state assumption, network topology and nutrient uptake rates [12]. This study presents an extended GEM of *T. cruzi*, hereafter referred to as iIS312, of the published and validated *T. cruzi* CL Brener core metabolism model iSR215 [13]. From the extended model iIS312, we built three stage-specific models using transcriptomics data [4] (Fig 1) generated from cultured parasites, as such models for other organisms (e.g., *Plasmodium falciparum*) have provided valuable insight of stage-specific changes in metabolism [10]. The resulting models of *T. cruzi* describe stage-specific differences in metabolism including pathway activity and essential genes.

## Results

### Reconstructing and expanding the metabolic network of *Trypanosoma cruzi*, iIS312

The reconstruction and curation of *T. cruzi* GEM (Fig 1, Step 1) was developed using the genome annotation of the strain Dm28c [14]. The curation was based on the two pre-existing validated models: iSR215 (*T. cruzi* strain CL Brener) [13] and iAC560 (*Leishmania major*

**a**

| Properties | Count |
|---|---|
| **Genes** | 312 |
| **Reactions** | 519 |
| **Gene associated (metabolic/transport)** | 328 (63%) |
| **No gene association (metabolic)** | 13 (2.5%) |
| **No gene association (transport)** | 132 (25%) |
| **Exchange** | 45 |
| **Demand (biomass)** | 1 |
| **Metabolites** | 606 |
| **Compartments** | 6 |

**b**

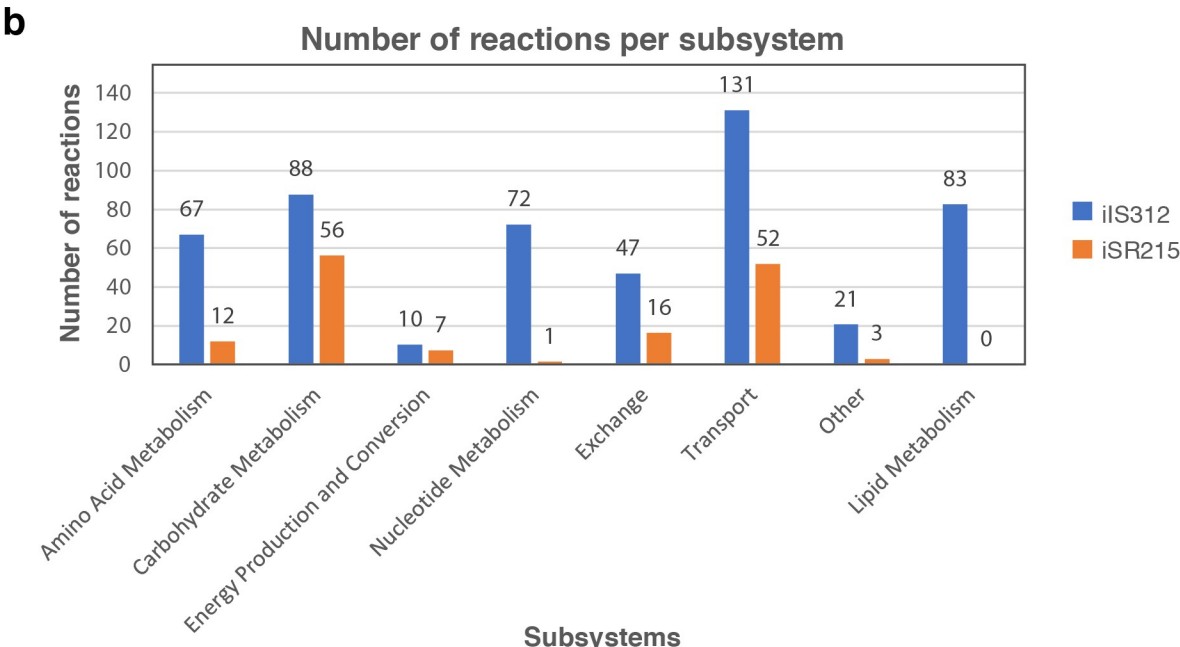

**Fig 2. Content description of iIS312.** (A) Properties of IS312, including number of genes, reactions, metabolites, and compartments. (B) Distribution of reactions across subsystems in iIS312 and iSR215.

strain Friedlin) [15]. The iAC560 model was the first metabolic network reconstruction built for a protozoan. In addition, *L. major* is a close-related organism to *T. cruzi*, as they both belong to the same family, *Trypanosomatidae*. The cross-referencing between genes in *T. cruzi* Dm28c and *T. cruzi* CL Brener and *L. major* Friedlin was carried out using the database Tri-TrypDB [16], which is a database specific for pathogens of the family *Trypanosomatidae* (*Leishmania* and *Trypanosoma* genera).

The expanded *T. cruzi* model iIS312 accounts for 519 reactions, 606 metabolites, 312 genes, and 6 compartments (Fig 2a, S1 Table), including the cytosol, mitochondria, glycosome, endoplasmic reticulum, nucleus, and the extracellular compartment. The extracellular compartment is included in order to simulate nutrient uptake through transport reactions from extracellular space. Among the 519 reactions, 328 (63%) have gene-association and 132 (27%) have no gene association (Fig 2a) due to a lack of genome annotation or exchange reactions that simulate nutrient uptake. Compared to the *T. cruzi* model iSR215, iIS312 significantly expanded the scope of the model (Fig 2b), including 83 new reactions in the lipid metabolism

that are absent in iSR215. Fig 2b indicates that the number of reactions increased for all subsystems in the updated model iIS312 (See S1 Fig).

In addition, we compared the essential reactions predicted by the model iIS312 with the experimental data used in the iSR215 work [13] (See S2 Table). iIS312 was able to reproduce 23 out of 29 results from essential gene experiments, which means that 79% of the gene essentiality model were accurate (See S2 Table).

## Stage-specific models generated through integration of transcriptomics data and validated by their predicted end products

Stage-specific models were generated from the pan model iIS312 and transcriptomics (S3 Table) by "deactivating" genes with low expression levels and their corresponding reactions. As indicated in S3 Table, out of 2,221 metabolic genes in the transcriptomics data, only 312 were included in our model after checking supportive evidence of their presence in *T. cruzi* metabolism in the literature. By integrating gene expression data with the pan iIS312 model, we generated three stage-specific models: Amastigote, Epimastigote, and Trypomastigote (see Fig 1, Step 3). During the integration of gene expression data, reactions were "deactivated" if all or some of their encoding genes (depending on the gene rule; see Methods) presented an expression level below a predefined threshold (see Methods). In addition, if the gene deactivation resulted in null-growth, gap-filling methods would take place to identify the genes that need to be reactivated to restore growth, and they would be kept active in the downstream analysis (see Methods). Taking the amastigote stage as an example, 68 genes should be deactivated according to the methodology, but only 59 genes (or 42 reactions) were kept deactivated as 9 of them were essential to restore growth to the model. Further discussion about it is presented in the next section.

The number of deactivated reactions differ for the three stage-specific models (Fig 3), showing consistency with literature findings regarding stage-specific metabolism activity. Out of all deactivated reactions, only one deleted reaction overlaps between the Amastigote and Trypomastigote models, suggesting substantial variation in metabolism between stages. The highest number of deactivated reactions was observed in the Trypomastigote model (70 reactions), followed by the Amastigote (42 reactions) and Epimastigote (7 reactions) models. Therefore, Trypomastigote is the stage with the least active metabolism, followed by amastigote and epimastigote, which is aligned with findings in the literature [4].

Literature suggests that the end products of *T. cruzi* metabolism include succinate [17–20], alanine [18,19], $CO_2$ [6], acetate [18,20], and glycine [21]. The potential of end products secretion was predicted by the iIS312 stage-specific models through flux variability analysis (FVA) [22], a method that is used to investigate the ranges of reaction fluxes for alternative optimal solutions. The prediction of end products fit the expected spectrum of metabolite secretion (Fig 3b and S4 Table), supporting the validity of the models (see Methods). We also compared the experimental observations with the predicted metabolite secretion by stage-specific models (details of stage-specific models discussed in the next section) whenever the literature evidence is available (Fig 3b). No literature evidence was available on glycine and CO2 for T. cruzi, we included them as they were found in a closely related organism T. brucei [21]. Although citrate showed up for the amastigote and epimastigote models, it was not included in Fig 3b due to lack of experimental data on it. The stage-comparison suggests that overall consistency is relatively high, with some minor inconsistencies, which could be potentially due to knowledge gaps and differences in experimental and simulated media. The excreted metabolites vary across stages, due to different constraints applied in each stage.

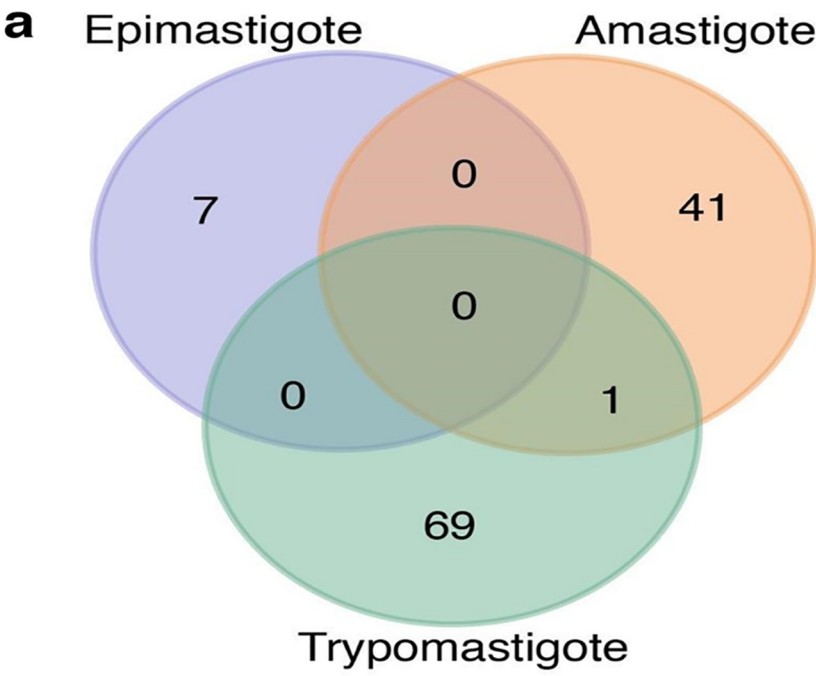

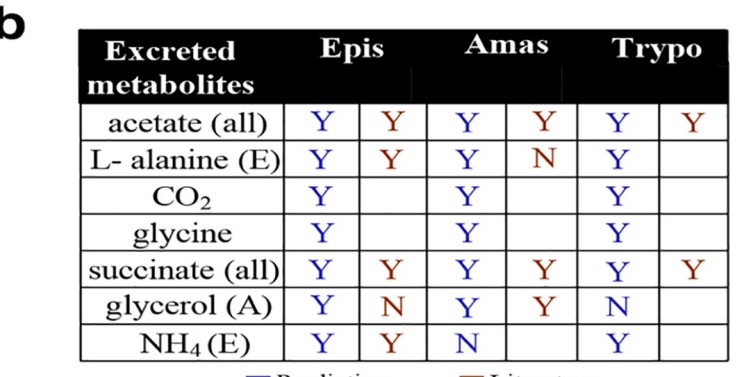

| Excreted metabolites | Epis | | Amas | | Trypo | |
|---|---|---|---|---|---|---|
| acetate (all) | Y | Y | Y | Y | Y | Y |
| L- alanine (E) | Y | Y | Y | N | Y | |
| $CO_2$ | Y | | Y | | Y | |
| glycine | Y | | Y | | Y | |
| succinate (all) | Y | Y | Y | Y | Y | Y |
| glycerol (A) | Y | N | Y | Y | N | |
| $NH_4$ (E) | Y | Y | N | | Y | |

■ Prediction     ■ Literature

**Fig 3. Construction and validation of stage-specific iIS312 models.** (A) Venn diagram of deactivated reactions for each stage-specific iIS312 model (see S6 Table). (B) Validation of stage-specific iIS312 models by their predicted end products (See S4 Table). Metabolites were marked with Y when secreted, N when it was not secreted, and blank when evidence is not available from the literature.

## Stage-specific models present significant differences among metabolic pathways

Variation in deactivated reactions across stages suggest stage-specific metabolic phenotypes. Most of the reactions "deactivated" in amastigotes (intracellular replicative stage) were involved in the TCA cycle, Pentose Phosphate Pathway (PPP), and Purine and Pyrimidine Metabolism. Literature suggests that the carbohydrate catabolism is down-regulated in amastigotes, while the transmembrane transport, macromolecule metabolism, and DNA replication are up-regulated [4]. The weak simulated metabolic flux through some TCA cycle reactions in the amastigote model (See Fig 4) might suggest that amastigotes optimize their growth by using alternative shortcuts in TCA (e.g., through amino acid metabolic pathways as suggested by a previous study) [7,23]. In addition, the deactivations in Purine and Pyrimidine

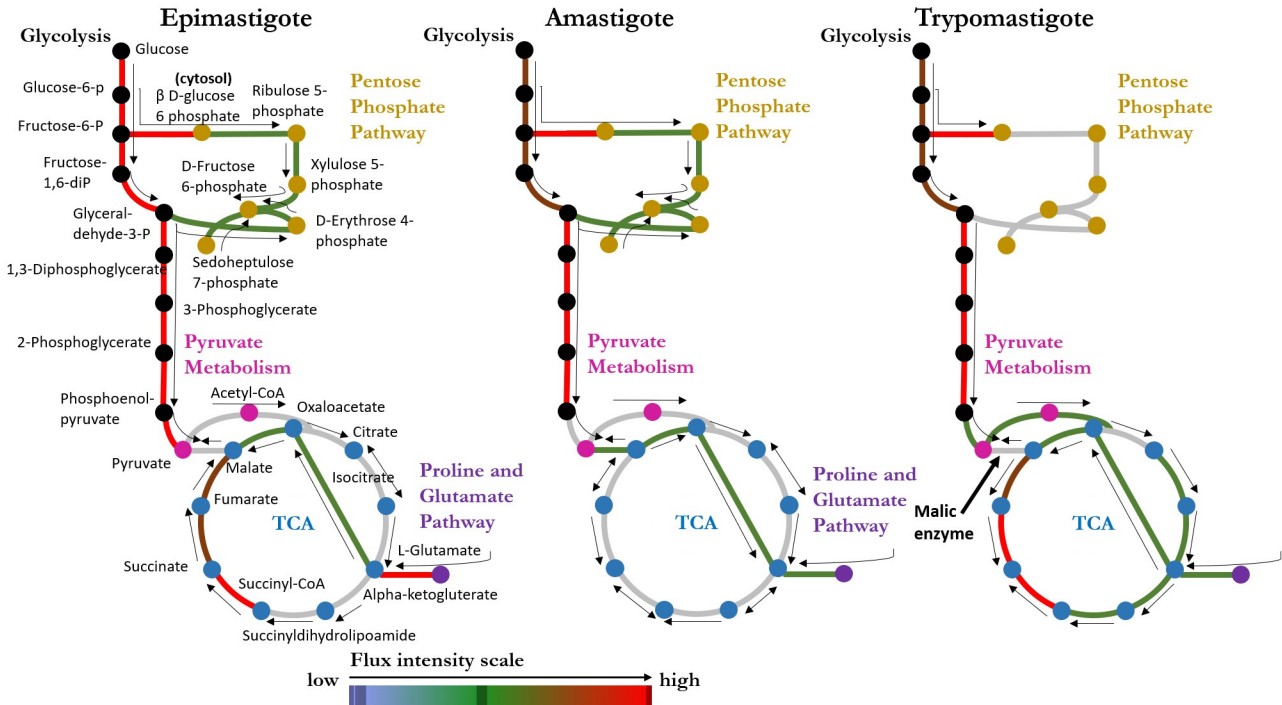

**Fig 4. Flux distribution representation of each *T. cruzi* stage model in central metabolism (Glycolysis, Pentose Phosphate Pathway, TCA, and others).** See S2, S3 and S4 Figs.

Metabolism were mostly associated with lyases, which might be related to the downregulation of nucleotide breakdown to favor DNA replication. In trypomastigotes (non-replicative and infective stage), the "deactivations" occurred in various subsystems, including Fatty Acid Biosynthesis, Glutamate metabolism, Glycerophospholipid metabolism, Glycolysis/Gluconeogenesis, Purine Metabolism, PPP, and Steroid Biosynthesis. Literature suggests that the nucleic acid metabolism is downregulated in trypomastigotes [4], which is consistent with the deactivated reactions from Purine Metabolism and PPP in our model. In epimastigotes (replicative and insect-specific stage), the literature also presented the downregulation of cell adhesion and no specific metabolic pathway [4]. Our model does not show any prevalent subsystem deactivation, as it only includes reactions associated with metabolic functions and not cell adhesion.

In particular, we also observed that various essential genes (genes encoding enzymes that are required for growth) identified in the pan model iIS312 were deactivated in stage-specific models, highlighting the significant changes in central metabolism across stages (see Fig 4). For amastigotes, ten of the deactivated genes were found as essential in the pan model iIS312 predictions, and they are enriched in Glycolysis/Gluconeogenesis (see S5 and S6 Tables), which is intriguing as the result suggests glycolysis is not essential for the amastigote stage. This is consistent with a previous finding that glucose transporter activity was not detected in the amastigote stage [23]. Therefore, it is likely that *T. cruzi* might have transporters for intermediate glycolytic metabolites [24,25], which explains the deactivation of reactions in Glycolysis in amastigotes. Confirmation of such transporters still needs further investigation in the future. For trypomastigotes, the deactivated genes that were found as essential in the pan model iIS312 are mostly involved in PPP, which is responsible for nucleic acid synthesis and the production of NADPH. This result is consistent with the literature, since trypomastigotes are a non-replicative stage, which does not require nucleic acids for DNA synthesis. Therefore,

we calibrated our biomass reaction for the trypomastigote model by removing metabolites involved in DNA synthesis (see S1 Text). For epimastigotes, the only deactivated gene that was found as essential in the pan model iIS312 is L-threonine dehydrogenase, that is responsible for threonine metabolism, suggesting that threonine may not be essential for parasite growth, or alternative pathways that are missing in the genome annotation are present.

In addition to deactivated reaction analysis, the flux distribution map for each stage-specific model was generated (see Methods) to compare the metabolic differences among stages (see S2, S3 and S4 Figs). A flux distribution map was generated using flux balance analysis [12] with the objective set as optimal growth (see Methods). As shown in Fig 4, the flux distribution in the central metabolism among the stages changed considerably. The reactions in gray are not necessarily inactive in the model, but they indicate weak simulated metabolic flux. While in the amastigote model, the flux in the TCA cycle takes a shortcut through the proline and glutamate pathway (entering the TCA cycle through alpha-ketoglutarate). In the trypomastigote model we have an opposite behavior, with almost all the TCA reactions being operative. The metabolic flux through the malic enzyme is only present in the amastigote model to optimize the pyruvate flux outwards the TCA cycle and produce alanine. In trypomastigotes, there is an apparent requirement for an active TCA for energy yield, potentially to energize the flagellar activity observed in these forms, allowing them to move around. The epimastigote model presented an intermediate behavior for the flux distribution, showing a TCA less utilized than trypomastigote but more active than amastigote. Flux variability analysis (FVA) of the stage-specific models also confirmed differences in the metabolic activity of *T. cruzi* stages (see more details in S7 Table and S5 Fig). S5 Fig shows the minimum and maximum flux in the central metabolism reactions for each stage, highlighting the differences in Glycolysis and Gluconeogenesis, TCA, PPP, Pyruvate Metabolism, and Glutamate and Proline Subsystem.

## Stage-specific models are experimentally validated

To test the accuracy of our model, we chose five enzymes involved in the core metabolism of the parasite: glucose-6-phosphate dehydrogenase, hexokinase, malic enzyme, glutamate dehydrogenase, and citrate synthase. To measure the activity of these enzymes, total cell-free protein extracts of each stage of the parasite were prepared. The same mass of total protein extracts was used to measure each enzyme activity, for all the stages (Fig 5). The selected enzymes are key points of metabolic pathways and some then showed to be differentially expressed in the *in silico* analyses. These enzymes were chosen due to their participation in critical metabolic processes such as glycolysis, TCA cycle, or the consumption of most of oxidizable amino acids. More specifically, hexokinase (HK) is the enzyme that catalyzes the first step of glycolysis, which produces glucose-6-phosphate. Noteworthy, the product of HK is as well a substrate for the first step of PPP. Glucose-6-phospate dehydrogenase (G6PDH), was chosen because it is the first enzyme of PPP. Remarkably, both showed opposite regulatory patterns of expression among proliferative and non-proliferative stages. Citrate synthase (CS) is a key enzyme of the TCA cycle, responsible for the entry of acetyl-CoA into this pathway, through its condensation with oxaloacetate. Oxaloacetate is an intermediate of the TCA cycle, produced by the enzyme fumarase. However, a malate dehydrogenase (oxaloacetate decarboxylating) or malic enzyme (ME), can short-circuit the main entry of pyruvate into the TCA cycle (which happens through the conversion of pyruvate into Acetyl-CoA) by feeding it through the production of oxaloacetate instead of citrate. Glutamate dehydrogenase was chosen because is one of the main links among amino acids metabolism and the TCA cycle in *T. cruzi*, by producing 2-oxoglutarate from glutamate.

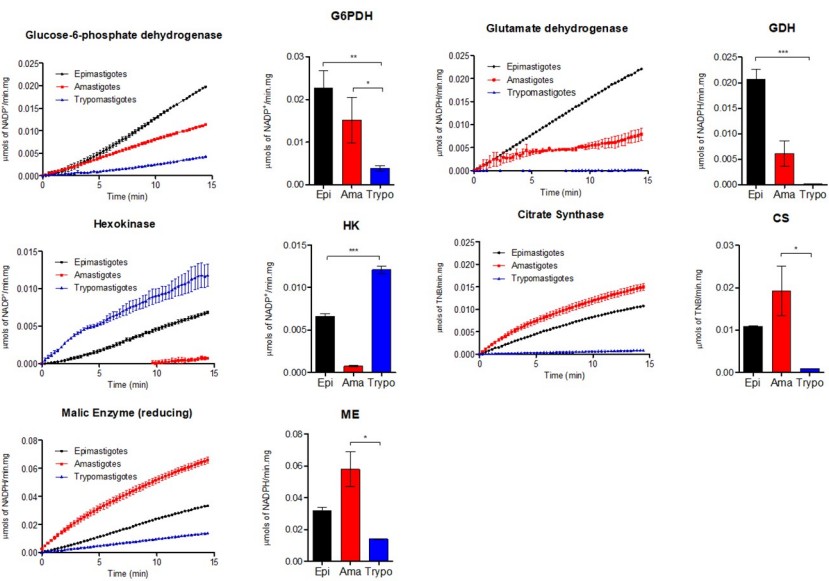

**Fig 5. Comparison of the activity of five enzymes involved in the core metabolism of *T. cruzi* in the three developmental stages of the parasite.** For each enzyme, the figure on the left shows the kinetics of the enzymatic activity for 15 minutes and the bar-charts on the right normalizes the activity per time to allow for a better comparison of activity in each stage of the parasite.

Our measurements showed that the model accurately predicts the comparative enzyme activity levels in the different stages of the parasite. The G6PDH activity was lower in the non-proliferative trypomastigote stage, being consistent with our model predictions and supporting the fact that PPP is diminished in non-proliferative forms. The HK activity was higher in epi-mastigotes and trypomastigote stages but showed low detectable levels of enzyme activity in the amastigote stage, not as predicted as shown in Fig 4. HK should have been deactivated in amastigote based on transcriptomics data, but was added back during the gap-filling process as its deletion disabled growth, possibly due to the unknown alternative pathways that are not present in the model. We expect to see consistency between HK flux prediction and measure-ments if such alternative pathways can be discovered and added to the model in the future. Nevertheless, this result is consistent with data in the literature showing a downregulation of glycolytic metabolism in amastigote forms due to down-regulation of the *T. cruzi* hexose trans-porter, which facilitates the entry of glucose into the cells [23]. Remarkably, an opposite regula-tion was found when HK and G6PDH activities were compared in the different stages. This is consistent with the model prediction that in amastigotes glycolysis is diminished, allowing a gluconeogenic flux to produce glucose-6-phosphate to feed PPP. The CS showed to be more active in the amastigote stage than in other stages, which is supported by an increase in the ME activity. ME expression in amastigote stage was previously reported [26]. Amastigotes forms apparently utilize this branch to fulfill TCA cycle, as expected based on the model predictions. GDH is more active in the epimastigote stage. This result is consistent, for example, with the pathways already described being responsible to convert amino acids such as proline [27,28], glutamine [29] and histidine [30] into glutamate, producing 2-oxoglutarate as a product. In addition, trypomastigotes shows low levels of GDH activity, which makes sense being the metabolism of these forms based in glucose consumption.

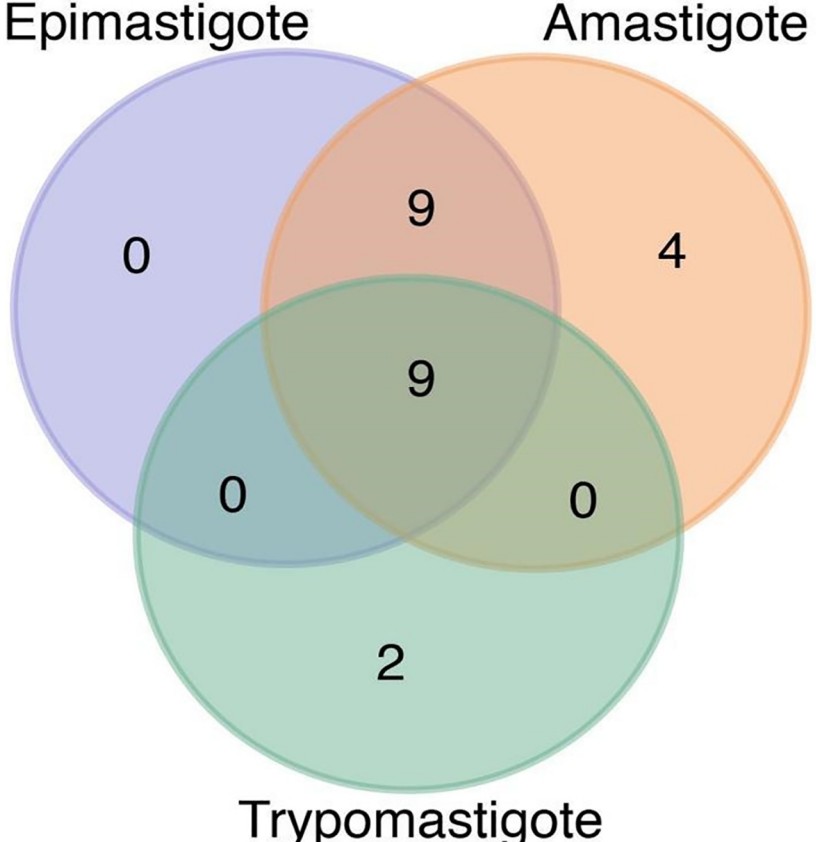

**Fig 6. Venn diagram of essential single genes for each stage-specific iIS312 model (see S9 Table).**

## Stage-specific differences in essential genes and metabolic flux suggest potential drug targets

Through growth simulation of stage-specific models, we identified essential genes and reactions that could potentially be drug targets. Single reaction and gene deletion simulations were performed (see Methods) for the stage-specific models built from transcriptomics integration. We generated a list of potential essential genes and reactions for *T. cruzi* growth at different stages (see S8 and S9 Tables), which can be useful for drug discovery for Chagas disease. Here we discuss the stage-specific essential genes identified, and their corresponding essential reactions can be found in S9 Table.

Genes that are essential for parasite survival differ for each stage, as the environment and objective change across stages. Stage-specific models predict that nine genes were essential for all *T. cruzi* stages (Fig 6), of which one belongs to the Glutamate Metabolism, two to Glycine, Serine, and Threonine Metabolism, and six to Glycolysis/Gluconeogenesis. We also found that nine essential genes were unique to epimastigote and amastigote (both replicative stages of *T. cruzi*), most of which belong to PPP, likely because trypomastigote has a different growth objective that does not require DNA replication. Finally, four essential genes were unique to amastigote and only two for trypomastigote. Some of the predicted essential genes have also been found as lethal for other studies in *T. cruzi*, e.g., glutamate dehydrogenase (encoded by TCDM_10134) [31], glucokinase and hexokinase [32]. Both hexokinase and other glycolytic enzymes can be tricky targets for the amastigote stage, since glycolysis may not be essential for

amastigote stage, as discussed in a previous section. Note that it is possible that the parasite could adapt and modify its metabolism under different conditions, which is out of the scope of this approach. We also performed double gene deletions and predicted lethal genes pairs that may potentially hinder cell survival if their products are simultaneously inhibited. This result provides potential drug targets that could be considered for Chagas disease treatment (see S10 Table). To select potential drug targets for further experimental validation, it is also important to minimize collateral damage to host cells, by focusing on enzymes that do not have homologs in the host [33], or enzymes that are essential to the pathogen but not for the host [34].

## Discussion

The development of drugs capable of targeting essential genes in multiple stages of *T. cruzi* might be the key for the improvement of Chagas disease treatment. In this work, we reconstructed the updated GEM of *T. cruzi*, iIS312, to incorporate the expansion of a recent genome annotation [14] of *T. cruzi*. Through integration of transcriptomics data, we generated three stage-specific models (Epimastigote, Amastigote, and Trypomastigote) and validated them against experimental observations in previous studies. The reconstruction of the stage-specific GEM allowed us to investigate the variation in metabolic functions across stages and predict stage-specific essential genes and reactions, which can be experimentally validated through the disruption of gene or gene products (e.g., drugs for enzyme inhibition and gene knockouts). Our predictions in gene and reaction essentiality, as well as variation in metabolic flux distribution across stages, may be a step forward towards the improvement of Chagas disease treatment. To our knowledge, these stage-specific models are the first GEMs built for the stages Amastigote and Trypomastigote. This work is also the first to present an *in silico* GEM comparison among different stages in the *T. cruzi* life cycle.

Not only is our model significantly expanded and improved compared to the existing *T. cruzi* model, the stage-specific GEMs allowed us to predict and compare the differences in metabolism across stages. Specifically, the stage-specific models suggest the potential non-essentiality of glycolysis in the amastigotes. This result is consistent with some previous findings including: 1) Silber et al. showed glucose transporter activity was not detected in amastigotes [23]; 2) Li et al. [24,25] showed that glycolytic gene expression levels drastically oscillated over time (0–72 hours post-infection—hpi)—glycolytic gene expression decreased during 0-6hpi when amastigotes infect mammalian cells, and then increased between 6 to 72 hpi, possibly due to the inability of the host to provide intermediate metabolites as it is being invaded by the parasite. While these findings reinforced the non-essentiality of glycolysis in amastigotes and suggest the existence of enzymes that are more important to sustain glycolytic flux in the pathogen than in the host, the essentiality of glycolysis in amastigotes is still being debated as other studies had different observations. We observed that the malic enzyme (ME) seems to play an important role in the amastigotes metabolism. ME is a NADPH dependent enzyme and was already found more expressed in amastigotes forms [26]. NADPH is important to support anabolic reactions such as the de novo synthesis of fatty acids [35], important for the building of glycosylphosphatidylinositol (GPI) molecules [36]. Due to the lower activity of G6PDH linked to the absence of glycolytic metabolism in these forms, the ME in amastigotes probably plays an important role in the supply of NADPH for the parasite.

Future work will involve the refinement of iIS312 through adjustments in the biomass reaction, better genome annotation and defined simulation media. Our eukaryote cell GEM presents 519 reactions, and the published 2009 model iSR215 only presents 162 reactions, while most recent prokaryotic cell GEMs present more than one thousand reactions [37]. The difference in the amount of reactions highlights the limitation in *T. cruzi* genome annotation and

how little we still know about this pathogen. Additionally, transcriptomics data may not be the most accurate representation of the protein level, especially given the transcriptomics data of amastigote stage used in this study is extracellular. Proteomics data may be a good alternative in the future to improve the accuracy. Another limitation of this work is that the transcriptomics data was collected from different complex growth media for different stages, yet we used the same simulation medium across stages, due to the lack of defined growth media for *T. cruzi*. This could potentially be addressed in the future with the development of defined media and corresponding stage-specific transcriptomics data to ensure consistency between simulation and actual growth. Additional future work also involves the construction of strain-specific *T. cruzi* models for a more accurate prediction of drug targets for different strains. It is important to note that the current GEM only delineates the metabolic capability of *T. cruzi*, without taking into account many other functions such as regulation, gene expression or infection. But this model will be a good starting point for building more comprehensive models such as a metabolic and gene expression model [38] and additional knowledge including structural information of metabolic enzymes [39] and regulatory functions can also be incorporated in the future.

## Materials and methods

### iIS312 metabolic network reconstruction

Multiple steps are involved in building expanding the metabolic network reconstruction of *T. cruzi*. The first step is curation of metabolic knowledge (Fig 1, Step 1), which consists of data collection from genome annotation and gene-protein-reaction databases, metabolic reaction list generation, and determination of gene-protein-reaction relations. To curate our model, we used the genome annotation of the strain Dm28c [14], TritrypDB [16], Uniprot [40], and the models iSR215 (core metabolism model reconstruction of *T. cruzi*) and iAC560 (genome-scale model reconstruction of *L. major*). Once the curation was accomplished (see S1 Table), the list of metabolic reactions was translated into a stoichiometric matrix (S), which is the mathematical representation of the stoichiometric coefficients of all substrates and products of all the reactions. Palsson et al (2016) [11] describe in their work how GEM can be useful to predict biological capabilities. Model reconstruction is performed using the toolbox COBRApy [41]. The model is named then following the rules introduced by Reed et al [42]: model name starts with 'i' to denote *in silico*, followed by the first author's first and last initials ('IS'), and by the number of genes in the model ('312'). The full metabolic network map can be found in the supplement (see Model Expansion, S1 Fig). We have also evaluated the iIS312 model via a model testing suit MEMOTE [43]. MEMOTE reported that the stoichiometric consistency, metabolite connectivity, mass and charge balance and metabolite and reaction identifier namespace to be >95%. The annotation information and sources are very specific for our model (as *T. cruzi* has its own database), therefore cannot be properly evaluated by MEMOTE. However, all annotation information of our model is available in S1 Table. Despite the annotation score, MEMOTE still reported an overall score of 81% for iIS312.

### Biomass reaction

The biomass reactions for *T. cruzi* was built based on the cellular composition of *L. major*, a related protozoan parasite. The function of the biomass equation is to provide a drain of essential metabolites that are needed to support the growth of the metabolic system [15,44,45]. Since the cellular composition is likely to vary according to the physiological conditions, the biomass equation is an approximation [44]. On the other hand, findings in the literature have shown that perturbations in the coefficients of metabolites in the biomass should not affect the

overall biomass yield significantly [44]. Biomass reaction is modified for trypomastigote model as it is the only non-replicative stage (see S1 Text). In addition, regarding the biomass consistency in the MEMOTE report, we found a value of 2.19 for the iIS312, amastigote and epimastigote models, and 1.64 for the trypomastigote model (as the biomass reaction for this stage was modified). Even though the values deviate from the ideal value 1, this result was expected due to limited data for the cellular composition of *T. cruzi*. For this reason, users should be cautious when comparing the growth rate of iIS312 and stage-specific models with other models.

## Growth simulation by flux balance analysis (FBA)

After the curation step and the construction of the stoichiometric matrix, the flux balance analysis (FBA) is applied in the model to simulate *T. cruzi* metabolism. FBA is a mathematical optimization problem whose goal is identifying a specific flux distribution that optimizes a given metabolic objective (in our case, the biomass reaction of the model). Palsson et al [12] discuss in their work what FBA is and its application in biochemical networks.

Since our model is simulating cell growth, the quasi-steady state $(S.v = 0)$ can be considered, as the time constants for this case are long (hours to days), which is different from the time constants for metabolic transients (<tens of seconds) [13,46].

The medium used for growth simulation for iIS312 was developed based on the ones used in *L. major* model iAC560 [15] and *T. cruzi* model iSR215 [21]. To define *in silico* growth conditions, we changed the lower bound of exchange reactions. For our FBA analysis, we allowed the intake and outtake of the following metabolites: glucose, $H^+$, phosphate, acetate, cysteine, alanine, arginine, asparagine, choline, citrate, aspartate, oxygen, succinate, glutamate, glycine, glycerol, threonine, carbon dioxide, proline, ammonium, deoxyribose, ergosterol, glutathione, guanine, hydrogen sulfide, histidine, hypoxanthine, isoleucine, methionine, nicotinamide, nicotinic acid, ammonia, phenylalanine, serine, thymine, uracil, and valine.

## Validation of the model expansion

The validation step (Fig 1 Step 2) consisted of comparing findings in the literature, enzyme assays and our model predictions. The findings we used to validate the model were the end products secreted by *T. cruzi*. We also compared our epimastigote model predictions to experiments involving targeted disruption of genes or gene products (gene knockouts, RNAi, and drugs for enzyme inhibition) in *T. cruzi* or related organisms (see S2 Table).

## Identification of essential genes

The essential gene-reactions list was generated by using the single gene deletion function in COBRApy [41], a python-based tool to build metabolic network reconstruction. The function was applied in the pan iIS312 model to identify essential genes in the pan model iIS312. Essential genes were also identified in stage-specific models. Specifically, for the amastigote model, we included the genes TCDM_04067 (ENO enolase, subsystem Glycolysis/Gluconeogenesis) and TCDM_05454 (PPCKg glycosomal phosphoenolpyruvate carboxykinase, subsystem Glycolysis/Gluconeogenesis) into the list of essential genes to maintain a non-nul growth.

## Transcriptomics integration and life cycle stage-specific models

We integrated transcriptomics data (Fig 1, Step 3) to build stage-specific models from the pan iIS312 model followed the method presented by Richelle et al [47]. The gene expression data

used came from the work of Berná et al [4], where the authors generated transcriptomics for all the stages in *T. cruzi* life cycle (epimastigotes, amastigotes and trypomastigotes).

We then used model extraction method (MEM) [47,48] to generate stage-specific models through integration of transcriptomics data. For our models, we used the MBA-like MEM [47–49], an algorithm that uses a set of core reactions that should not be removed, while removing other reactions with low gene expression. We defined the set of essential gene-reactions obtained from the pan model iIS312 as the set of core reactions that should not be removed. For the preprocessing of gene expression data, we attributed a gene activity score for each gene and defined a threshold to determine which genes are active in each *T. cruzi* stage. The gene threshold is usually defined by the mean of each gene expression level over all the sample stages coming from the same dataset. In addition, the threshold should be higher or equal to the 25[th] percentile of the overall gene expression value distribution and lower or equal to the 75[th] percentile [47]. The gene score is given by:

$$Gene\ score = 5\ log\ \left(1 + \frac{Expression\ Level}{Threshold}\right)$$

The gene scores are integrated into the model by parsing the Gene-protein-reaction association (GPR) rules (see S1 Table) associated with each reaction. Since the GPR is a logical expression (i.e. "AND" and "OR" logical operators), the gene score for each reaction is defined by tacking the minimum expression value among all the genes associated to an enzyme complex ("AND" rule) and the maximum expression value among all the genes associated to an isozyme ("OR" rule) [47,50].

As stated previously, our MEM considers a set of essential gene-reactions that should not be deactivated. The gene scored of each gene was calculated using the given formula above. The choice of the threshold in the gene score formula was defined first as the 25[th] percentile of the overall gene expression value distribution, given the variability in the transcriptomics data of *T. cruzi* stages. The gene scores were then compared to another threshold equals to 5log2, as recommended in the literature [47]. For each gene, we evaluated if the gene score was lower than 5log2 and if it belonged to the essential gene-reactions list. If these two conditions were satisfied, the gene was knockout in the model.

During the generation of stage-specific models, some essential genes were deactivated following the above procedure due to low gene-expression levels. This is likely because they are not true essential genes. They were predicted to be essential because there are alternative metabolic pathways that has not been discovered and therefore not included in this model. In order to restore growth of stage-specific models and perform meaningful analysis, we used gap-filling function in COBRApy [41] to identify the reactions/genes that needs to be reactivated to restore growth, and kept them active in the downstream analysis. The tool was able to predict which reactions were missing to enable growth in the amastigote model, which includes two from Glycolysis/Gluconeogenesis TCDM_04067 (ENO enolase) and TCDM_05454 (PPCKg glycosomal phosphoenolpyruvate carboxykinase).

## Simulation of end product secretion profiles

The end products were generated by FVA in COBRApy [41]. The biomass reaction was set up as the objective function. The metabolite is considered secreted if the maximum possible flux through the fermentation product exchange reaction was positive according to a previous study [51]. The secretion profile for this analysis is shown as presence/absence on S5 Table.

## Flux distribution maps, metabolic network and gene-expression maps and venn diagrams

The flux distribution maps were generated by Escher-FBA [52]. This tool is a web application for interactive flux balance analysis (FBA) simulations within a pathway visualization. We uploaded our metabolic network maps and models to generate the output. In addition, all metabolic network and gene-expression maps were generated by Escher [53]. This tool is also a web application for building and generating data-rich visualizations for biological pathways. We also integrated transcriptomics into Escher map to facilitate the identification of trends in gene expression data.

## Biochemical activity of core metabolism enzymes to verify the accuracy of the model

**Parasites.**    *T. cruzi* (strain CL, clone 14) epimastigotes forms were maintained in exponential growth, by cultivating them in LIT (Liver Infusion Tryptose) medium supplemented with 10% fetal calf serum (FCS) and 0.2% glucose at 28 ˚C (Camargo 1964). The infective trypomastigote stages were obtained from infections of CHO-$K_1$ cells (Chinese Hamster Ovary) as previously described [29,54]. Amastigotes forms were collected by lysing the CHO-$K_1$ cells 48 hours post-infection, as described in [29].

**Extracts preparation.**    The parasite (all stages) were washed twice in PBS and the pellets were resuspended in lysis buffer (Tris-HCl 20 mM pH 7.9, EDTA 1 mM, Sucrose 250 mM and Triton X-100 0.1%) supplemented with a protease inhibitors cocktail (Sigma) according to the manufacturer's instructions. The extracts were obtained by freeze—thaw in liquid $N_2$. The lysates were centrifuged at 10,000 x g, 4 ˚C for clarification, and the soluble fraction recovered and quantified by Bradford. Importantly, all the measured enzymes have been described as soluble proteins.

**Enzyme assays.**    All the methods for measuring the enzyme activities were established for being measured spectrophotometrically. For reading the assays we used a SpectraMax i3 (Molecular Devices, Sunnyvale, CA, USA) plate reader, at 28 ˚C, using 0.015 mg of total soluble fractions from each form in a final volume of 0.2 mL. All the measurements were performed at least in biological triplicates, each one measured at least in triplicates (technical triplicates). Data represent the average value from biological triplicates.

Hexokinase (HK): The activity was measured coupling the hexokinase activity with the commercial G6PD, following the reduction of $NADP^+$ at 340 nm. The reaction buffer contained 50 mM Triethanolamine buffer pH 7.5, 5 mM $MgCl_2$, 100 mM KCl, 10 mM glucose, 5 mM ATP and 5 U of commercial G6PD (Sigma) (adapted from [55]).

Glucose-6-phosphate dehydrogenase (G6PD): The activity was measured in the direction of G6P formation by $NADP^+$ reduction at 340 nm. The reaction buffer contained 50 mM Triethanolamine buffer pH 7.5, 5 mM MgCl2, 1 mM G6P and 0.5 mM $NADP^+$ [56].

Malic enzyme (ME): The activity was measured in the direction of L-malate decarboxylation, by the reduction of $NADP^+$ at 340 nm. The reaction buffer contained 50 mM Tris-HCl pH 7.5, 9 mM L-malate, 0.15 mM $NADP^+$, 2 mM $MnCl_2$ [57].

Citrate synthase: The activity was measured by DTNB reduction at 412 nm. The reaction buffer contained 100 mM Tris-HCl pH 8, 0.1 mM DTNB, 0,3 mM acetyl-CoA and 0.5 mM oxaloacetate [58].

Glutamate dehydrogenase: The activity was measured in the direction of glutamate oxidation, followed by the reduction of $NADP^+$ at 340 nm. The reaction buffer contained 100 mM Tris-HCl pH 7.5, 2.5 mM $NADP^+$ and 10 mM glutamate [59].

## Supporting information

**S1 Text. Supplementary methods.**
(DOCX)

**S1 GEM. Genome-scale metabolic model of *T. cruzi* amastigote stage in ".json" format.**
(JSON)

**S2 GEM. Genome-scale metabolic model of *T. cruzi* epimastigote stage in ".json" format.**
(JSON)

**S3 GEM. Genome-scale metabolic model of *T. cruzi* trypomastigote stage in ".json" format.**
(JSON)

**S4 GEM. Genome-scale metabolic models of all stages of *T. cruzi* in ".json" format.**
(JSON)

**S1 Table. Model content of iIS312.**
(XLSX)

**S2 Table. Comparison of model predictions and experimental data for lethal reactions in *T. cruzi* and related organisms.**
(XLSX)

**S3 Table. Transcriptomics data and integration.**
(XLSX)

**S4 Table. Excreted metabolites prediciton.**
(XLSX)

**S5 Table. List of lowly expressed essential genes across *T. cruzi* stages.**
(XLSX)

**S6 Table. List of stage-specific deactivated reactions after transcriptomics integration.**
(XLSX)

**S7 Table. FVA and pFBA simulation results.**
(XLSX)

**S8 Table. List of essential reactions after single reaction deletion.**
(XLSX)

**S9 Table. List of essential genes after single gene deletion.**
(XLSX)

**S10 Table. List of essential pairs of genes after double gene deletion.**
(XLSX)

**S1 Fig. Model expansion iIS312.**
(PDF)

**S2 Fig. FBA epimastigote.**
(PDF)

**S3 Fig. FBA amastigote.**
(PDF)

**S4 Fig. FBA trypomastigote.**
(PDF)

**S5 Fig. FVA central metabolism.**
(JPG)

## Author Contributions

**Conceptualization:** Isabel S. Shiratsubaki, Xin Fang, Bernhard O. Palsson, Jair L. Siqueira-Neto.

**Data curation:** Isabel S. Shiratsubaki, Xin Fang, Rodolpho O. O. Souza, Ariel M. Silber, Jair L. Siqueira-Neto.

**Formal analysis:** Isabel S. Shiratsubaki, Xin Fang, Rodolpho O. O. Souza.

**Funding acquisition:** Bernhard O. Palsson, Ariel M. Silber, Jair L. Siqueira-Neto.

**Investigation:** Isabel S. Shiratsubaki, Xin Fang, Rodolpho O. O. Souza, Ariel M. Silber, Jair L. Siqueira-Neto.

**Methodology:** Isabel S. Shiratsubaki, Xin Fang, Rodolpho O. O. Souza.

**Project administration:** Jair L. Siqueira-Neto.

**Resources:** Bernhard O. Palsson, Ariel M. Silber, Jair L. Siqueira-Neto.

**Software:** Bernhard O. Palsson.

**Supervision:** Ariel M. Silber, Jair L. Siqueira-Neto.

**Validation:** Isabel S. Shiratsubaki, Xin Fang, Rodolpho O. O. Souza, Ariel M. Silber, Jair L. Siqueira-Neto.

**Visualization:** Isabel S. Shiratsubaki, Xin Fang, Rodolpho O. O. Souza, Ariel M. Silber, Jair L. Siqueira-Neto.

**Writing – original draft:** Isabel S. Shiratsubaki, Xin Fang, Rodolpho O. O. Souza, Ariel M. Silber, Jair L. Siqueira-Neto.

**Writing – review & editing:** Isabel S. Shiratsubaki, Xin Fang, Rodolpho O. O. Souza, Ariel M. Silber, Jair L. Siqueira-Neto.

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
