## [Decision Letter · Decision Letter 0]

17 Apr 2020

Dear Dr. Siqueira-Neto,

Thank you very much for submitting your manuscript "Genome-scale metabolic models highlight stage-specific differences in essential metabolic pathways in Trypanosoma cruzi" for consideration at PLOS Neglected Tropical Diseases. As with all papers reviewed by the journal, your manuscript was reviewed by members of the editorial board and by several independent reviewers. In light of the reviews (below this email), we would like to invite the resubmission of a significantly-revised version that takes into account the reviewers' comments. 

We cannot make any decision about publication until we have seen the revised manuscript and your response to the reviewers' comments. Your revised manuscript is also likely to be sent to reviewers for further evaluation.

Sincerely,

Igor C. Almeida

Associate Editor

Joseph Vinetz

Deputy Editor

Reviewer's Responses to Questions

**Key Review Criteria Required for Acceptance?**

**Methods**

-Are the objectives of the study clearly articulated with a clear testable hypothesis stated?

-Is the study design appropriate to address the stated objectives?

-Is the population clearly described and appropriate for the hypothesis being tested?

-Is the sample size sufficient to ensure adequate power to address the hypothesis being tested?

-Were correct statistical analysis used to support conclusions?

-Are there concerns about ethical or regulatory requirements being met?

Reviewer #1: see summary and general comments section

Reviewer #2: The objectives are clear. and methodology is in teh direction of objectives. Authors develop a computational approach, and confrim their results through biochemical validation.

**Results**

-Does the analysis presented match the analysis plan?

-Are the results clearly and completely presented?

-Are the figures (Tables, Images) of sufficient quality for clarity?

Reviewer #1: see summary and general comments section

Reviewer #2: Results are complete, original, and Figures, Tables and Supplementary information allow to follow them clearly.

**Conclusions**

-Are the conclusions supported by the data presented?

-Are the limitations of analysis clearly described?

-Do the authors discuss how these data can be helpful to advance our understanding of the topic under study?

-Is public health relevance addressed?

Reviewer #1: see summary and general comments section

Reviewer #2: The conclusions are supported by the results, and authors discuss how these data can be helpful in the identification of targets and development of drugs against T. cruzi, causative agent of Chagas disease, a neglected disease of relevance in public health.

**Editorial and Data Presentation Modifications?**

Reviewer #1: see summary and general comments section

Reviewer #2: Minor Revision

**Summary and General Comments**

Reviewer #1: The manuscript by Shiratsubaki et al. describes the reconstruction of an updated genome scale metabolic model of Trypanosoma cruzi, iIS3142. The authors used the expansion of the genome annotation of T. cruzi to extend a previous model of core metabolism iSR215, while using also information from the Leishmania model iAC560. Using transcriptome data for the different lifecycle stages of T. cruzi, they made lifestage specific models. This allowed the authors to compare the metabolic networks in the lifestages and how metabolism sustains growth. They validated some of the network differences by measuring enzyme activities in the different lifestages.

This is a very relevant study as this extends the knowledge on the metabolic capabilities of T. cruzi. Metabolic enzymes could be potential drug targets and models can help in identifying the most promising drug targets in metabolism. 

While it is an interesting paper, there are a few major points that I would like to see addressed – also to make it more accessible to a non-GEM/FBA readership

Major points

1. Although there is an intrinsic value in extending the previous model, it is not clear to me whether the new reconstruction improved predictions or not. The authors should make a comparison of the prediction capabilities between both models and comment why the new model is better or worse. For example, for the genes which are common to both models, how many of them are accurately predicted to be essential using both models.

2. I wonder why the authors considered the biomass objective as the function to optimize for all the stages. This make sense for epimastigotes and amastigotes that replicate. However, it does not make sense for trypomastigotes which do not replicate. Non-replicating cells need to produce considerably less compounds, mainly for maintenance and pathogenicity and it seems very unlikely that the cell is trying to optimize biomass yield as they don’t replicate. Although the authors modified the biomass equations as stated in the supplementary material methods, it seems that this change is insufficient. For example, the new biomass equation still considers the same coefficients for amino acids. It is highly unlikely that replicating and non-replicating cell need the same amount of amino acids for protein synthesis. Please comment on this.

3. Despite that the model was validated regarding by-products and reaction essentiality, the model should also be validated in other aspects too to ensure a basic functioning and usability. Recently, Memote, a suite to assist in the assessment of genome-scale models, was published. We strongly suggest the authors to use this suite to further validate the model and include the results in the manuscript (see: https://doi.org/10.1038/s41587-020-0446-y)

4. It is not clear to me how big is the non-metabolic effect regarding inactivation of genes. I would like to have a bigger picture regarding transcriptomics and the genome-scale model. For example, how many genes overlap between the transcriptomics and the genome-scale model? How many genes are present in the transcriptomics and missing in the model? How many genes were deactivated according to the transcriptomics but they cannot be taken into account because they are not in the model? Do the authors expect only metabolic (and not regulatory) changes among the different stages?

5. The authors at several points in the manuscript state that metabolism could yield potential drug targets. While I agree with this, many metabolic pathways and enzymes are also operational in the human host. Therefore, the authors should discuss how side-effects could be prevented (e.g. because some pathways or enzymes may be less important in host metabolism)(see e.g. https://doi.org/10.1038/msb.2010.115 and https://doi.org/10.1038/srep40406)

6. It seems that the authors calculated the results of Fig 2C using parsimonious FBA. This is not fully correct to explore the potential of a metabolic network regarding products. Flux variability analysis should be performed to explore which metabolic products can be produced at optimal state. Also, it would be interesting to see what products must be produced at optimal state. For a reference regarding studying the potential for metabolic products, see https://doi.org/10.1038/nbt.3703

7. Why is the Pan-model part of Fig 2C? This model is not biologically relevant as this is not a lifecycle stage that exists. It could, theoretically, be interesting to show the full metabolic potential of T. cruzi. But a flux analysis is a functional analysis and should be restricted to the actual lifestages. This comment is also relevant for the analysis of PAN for deletions.

In addition, it would be more informative if Fig 2C could compare the FBA outcome per lifestage to what is known of the excreted products for the different lifestages from literature.

Minor points

a. Line 119 and Fig 2a: Please explain why an extracellular compartment is part of a GEM of a cell. This may be logic to modellers, but a short explanation would make the paper wider accessible.

b. The manuscript often refers to metabolic BYproducts. This suggests that there a more relevant purpose of metabolism (making growth possible!), but the authors never make this explicit in the manuscript

c. At line 128, FBA is first mentioned. While the methods section explains it further and provide useful references, for the less-informed reader we would like to see a short overall explanation of what is the purpose of this method and what type of questions this approach will answer. This should include the objective function that is used (a biomass equation) and the quasi-steady state assumption. The authors can keep it short, but it should convey the essence of the approach.

d. The legend to Fig 2C should be extended with explain what the PAN model is (if it stays in Fig 2C) and why these excreted metabolism are listed in the model (previous experimental data?)

e. Fig 3: There are no reactions that are deactivated in all lifestages. Was this expected?

f. Fig 4: It would be helpful to include substrates, end-products and directions of flux. In the amastigotes, there is no substantial flux from pyruvate to TCA, so the substrate for lower part are the amino acids, but this does not become immediately clear from the figure. Especially as line 269 mentions a gluconeogenic flux.

g. Fig 5: Please make the order of the lifecycle stages the same as Fig 4 for easy reference.

h. Fig 5: How is hexokinase a validation of the models? It has a high flux intensity in all models according to figure 4.

i. Line 391: the medium that was used for growth simulation was based on earlier in silico studies? Why didn’t the authors use the composition of the medium that was used in the validation experiments. 

j. Line 421: Does MEM include gap-filling? Or gap-filling was performed after MEM?

k. Lines 447-456: The authors say that some essential genes were deactivated in the model due to low gene-expression levels. Therefore, they removed a few reactions which made the model infeasible. To restore growth, they then do gap-filling using the template model. Why they didn’t used the BiGG database for the gap-filling procedure? In this way they can expand the search for alternative pathways as they suggest

Reviewer #2: In this manuscript, authors a genome scale metabolic model with focus on targets for development of drugs against Chagas disease, a neglected disease, in which currently two drugs are used -Nifurtimox nd Benznidazol- with several side/undesirable effects. They take into account the expression in the different stages, extracted from previously published transcriptomic studies. Particularly, they describe metabolic differences among stages, some of them already validated in previous publications, and by measuring the activity of the enzymes G-6-P dehydrogenase, malic enzyme, hexokinase, glutamate dehydrogenase and citrate synthase, confirming their predicted results. 

I consider that this manuscript gives very relevant information about metabolic fluxes in the relevant stages in mammals (amastigotes and trypomastigotes), which gives important clues about valid drug targets in T. cruzi. In my opinion this paper should be accepted after minor changes:

- Figure 1, "epimastigote" (instead of "espimastigote")

- Line: 191: "amastigotes" (instead of "amastigote")

- authors should consider to discuss about the role of malic enzymes in amastigotes, i.e.: anaplerotic reaction vs. NADPH source, taking into acount that PPP is not so active at this stage.

PLOS authors have the option to publish the peer review history of their article (what does this mean?). If published, this will include your full peer review and any attached files.

Reviewer #1: No

Reviewer #2: No
---

## [Decision Letter · Decision Letter 1]

15 Jul 2020

Dear Dr. Siqueira-Neto,

Thank you very much for submitting your manuscript "Genome-scale metabolic models highlight stage-specific differences in essential metabolic pathways in Trypanosoma cruzi" for consideration at PLOS Neglected Tropical Diseases. As with all papers reviewed by the journal, your manuscript was reviewed by members of the editorial board and by several independent reviewers. The reviewers appreciated the attention to an important topic. Based on the reviews, we are likely to accept this manuscript for publication, providing that you modify the manuscript according to the review recommendations. 

Sincerely,

Igor C. Almeida

Associate Editor

Joseph Vinetz

Deputy Editor

Reviewer's Responses to Questions

**Key Review Criteria Required for Acceptance?**

**Methods**

-Are the objectives of the study clearly articulated with a clear testable hypothesis stated?

-Is the study design appropriate to address the stated objectives?

-Is the population clearly described and appropriate for the hypothesis being tested?

-Is the sample size sufficient to ensure adequate power to address the hypothesis being tested?

-Were correct statistical analysis used to support conclusions?

-Are there concerns about ethical or regulatory requirements being met?

Reviewer #1: yes

Reviewer #2: (No Response)

**Results**

-Does the analysis presented match the analysis plan?

-Are the results clearly and completely presented?

-Are the figures (Tables, Images) of sufficient quality for clarity?

Reviewer #1: yes

Reviewer #2: (No Response)

**Conclusions**

-Are the conclusions supported by the data presented?

-Are the limitations of analysis clearly described?

-Do the authors discuss how these data can be helpful to advance our understanding of the topic under study?

-Is public health relevance addressed?

Reviewer #1: yes

Reviewer #2: (No Response)

**Editorial and Data Presentation Modifications?**

Reviewer #1: Panel 2C may go to Fig3

Reviewer #2: (No Response)

**Summary and General Comments**

Reviewer #1: Many thanks to the authors for the answers to the points. The manuscript has greatly approved in the revision. 

There are a few remaining points on the revised version. Where relevant, there is a reference to the lines in the Word document that contained the track changes (with the track changes on):

1. Considering the response to our first point: please indicate in the main text that only 23 genes could be validated (due to limited experimental studies) and that for those genes 79% of the predictions was accurate. 

2. Considering the response to our third point: can the authors say something on the MEMOTE results with respect to biomass consistency?

3. Considering the response to our fourth point: A very good comparison – we advise to put the most relevant outcomes in the main text

4. It is confusing that at line 135 and Fig 2C the stage-specific models are already used, while the text on how they are made comes only at line 162. Fig 2C could go to Fig 3 and the text about these results too?

5. The sentence at line 163 is superfluous as lines 165-169 say the same, but are more informative

6. At line 205 there is still a reference to the pan-model. Maybe include the number of the reconstruction as this makes it more clear what the pan model means

7. Line 218 suggests that only one reaction is deactivated in epimastigotes, but according to Fig 3 this should be seven reactions.

8. Line 332/333: a better formulation would be: “…, or enzymes that are more important to sustain glycolytic flux in the pathogen than in the host”. 

9. At line 354-357, please write dpi (days post induction??) in full

Reviewer #2: (No Response)

PLOS authors have the option to publish the peer review history of their article (what does this mean?). If published, this will include your full peer review and any attached files.

Reviewer #1: Yes: JR Haanstra

Reviewer #2: No
---

## [Editor Report · Decision Letter 2]

17 Aug 2020

Dear Dr. Siqueira-Neto,

We are pleased to inform you that your manuscript 'Genome-scale metabolic models highlight stage-specific differences in essential metabolic pathways in Trypanosoma cruzi' has been provisionally accepted for publication in PLOS Neglected Tropical Diseases.

Best regards,

Igor C. Almeida

Associate Editor

Joseph Vinetz

Deputy Editor

---

## [Editor Report · Acceptance letter]

29 Sep 2020

Dear Dr. Siqueira-Neto,

We are delighted to inform you that your manuscript, "Genome-scale metabolic models highlight stage-specific differences in essential metabolic pathways in Trypanosoma cruzi," has been formally accepted for publication in PLOS Neglected Tropical Diseases.

Best regards,

Shaden Kamhawi

co-Editor-in-Chief

Paul Brindley

co-Editor-in-Chief
